# Risk Profiling and Modulation for LLMs

## Abstract

Large language models (LLMs) are increasingly used for decision-making tasks under uncertainty; however, their risk profiles and how they are influenced by prompting and alignment methods remain underexplored. Existing studies have primarily examined personality prompting or multi-agent interactions, leaving open the question of how post-training influences the risk behavior of LLMs. In this work, we propose a new pipeline for eliciting, steering, and modulating LLMs' risk profiles, drawing on tools from behavioral economics and finance. Using utility-theoretic models, we compare pre-trained, instruction-tuned, and RLHF-aligned LLMs, and find that while instruction-tuned models exhibit behaviors consistent with some standard utility formulations, pre-trained and RLHF-aligned models deviate more from any utility models fitted. We further evaluate modulation strategies, including prompt engineering, in-context learning, and post-training, and show that post-training provides the most stable and effective modulation of risk preference. Our findings provide insights into the risk profiles of different classes and stages of LLMs and demonstrate how post-training modulates these profiles, laying the groundwork for future research on behavioral alignment and risk-aware LLM design.

## 1 Introduction

Large language models (LLMs) have rapidly advanced in capability and are increasingly being considered for domains where decision-making under uncertainty is central, such as financial services (Ding et al., 2024; Okpala et al., 2025), healthcare (Rao et al., 2023; Shool et al., 2025), and education (Wen et al., 2024; Chu et al., 2025). While their reasoning and language generation abilities are well established, recent work has begun to probe whether LLMs exhibit systematic behavioral tendencies similar to those documented in human decision-making (Kamruzzaman & Kim, 2024; Suzuki & Arita, 2024; Wang et al., 2025). For example, studies have shown that LLMs may display forms of risk aversion, probability weighting, or loss aversion reminiscent of prospect theory (Hartley et al., 2025), while others highlight divergences that make them appear more rational, or less consistent, than humans (Ross et al., 2024). Related research has explored how factors such as personality prompting, socio-demographic embedding, or role-playing interventions can systematically shift an LLM's preferences, suggesting that these models encode latent decision-making patterns that can be elicited or modified (Jiang et al., 2023; Jia et al., 2024).

Despite these developments, an important dimension remains underexplored: how post-training (Kumar et al., 2025) methods such as fine-tuning and reinforcement learning from human feedback (RLHF) shape the risk profiles of LLMs. These methods are now standard practice (Ouyang et al., 2022; Guo et al., 2025) for aligning LLMs with human preferences, improving safety, and tailoring behavior to application-specific needs. However, their implications for risk-related decision-making remain less understood. While post-training better adapts models to specific tasks (Guo et al., 2017), it can also introduce problems including knowledge forgetting, overfitting to evaluation benchmarks, and other safety concerns (Qi et al., 2023; Sun & Dredze, 2024). Therefore, in the context of risk evaluation and risk modulation, it is crucial to understand how different models behave under various training paradigms. Post-training may attenuate or amplify tendencies such as risk aversion, alter sensitivity to losses, or induce new forms of behavioral consistency across tasks. Given the growing demand for post-trained open-source models in real-world decision-making systems (Wang et al., 2023; Buckley et al., 2025), understanding these effects becomes crucial.

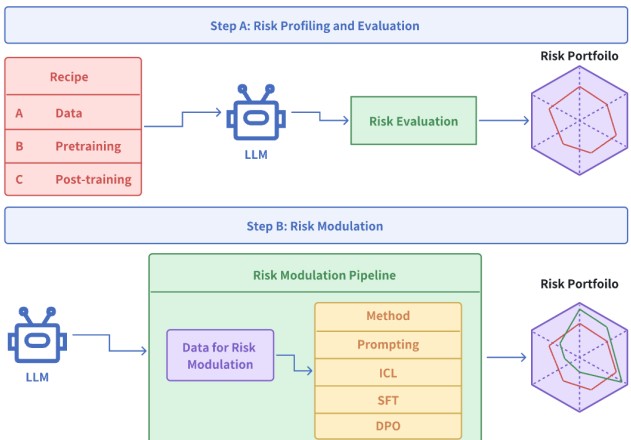

Figure 1: The general pipeline of our paper. In Section 2 and Section 3, we study the problem of profiling and evaluating the risk behavior of LLMs. In Section 4, we study several approaches in modulating the LLM's risk preference.

In this work, we propose a systematic pipeline to evaluate the impact of post-training on LLM risk-related behavior. Our approach draws inspiration from behavioral economics and personal finance, where human decision-making has long been studied through parameters such as risk preference and loss aversion (Tversky & Kahneman, 1992; Kahneman & Tversky, 2013; Von Neumann & Morgenstern, 2007). By adapting these evaluation tools to LLMs, we are able to quantify how fine-tuning and RLHF reshape models' tolerance toward uncertainty. Thus, our framework enables the *modulation* of *risk profiles* for LLMs, capturing how different post-training strategies modulate preferences toward gains, losses, and probabilities. This allows us to move from qualitative observations to quantitative characterization of risk-related behavior, paving the way for systematic comparison across models and alignment techniques. Our analysis shows that open-source LLMs, once post-trained, exhibit more consistent and reproducible patterns of risk tolerance compared to their pre-trained counterparts (Ross et al., 2024; Hartley et al., 2025). We further demonstrate that these patterns are not only more stable than those elicited by prompting, but also reveal structured ways in which post-training can modulate risk profiles.

Our contributions are summarized as follows:

**Pipeline for risk preference**. We introduce a new pipeline for eliciting, steering, and modulating LLMs' risk-related behavior. In particular, we investigate how post-training with multiple-choice tasks (involving financial, economic, and operational risks) can adjust models' risk preference.

**Different risk profiles for pre-trained and post-trained LLMs**: We fit a range of utility models to capture the risk profiles of both pre-trained and post-trained LLMs. Our analysis shows that instruction-tuned models exhibit behaviors consistent with standard utility formulations, whereas pre-trained and RLHF-aligned models deviate from all utility models considered. Interestingly, the best-fitting utility functions are often non-concave, which contrasts with the classical assumption of diminishing marginal utility. Our analyses provide a tool to visualize and interpret the implicit decision rules guiding LLM behavior.

**Modulation across pre-trained and post-trained LLMs**: We study three different approaches to modulate the risk preference of the LLMs – (i) prompt engineering, (ii) in-context learning, and (iii) post-training. We find the post-training approach can effectively modulate the risk-preference for different LLMs. The prompt engineering brings qualitative rather than quantitative adjustment, while the in-context learning turns out to be unstable.

We defer more discussions on related works to Appendix A.

# 2 An Initial Risk Assessment of LLMs

## 2.1 Grable & Lytton risk tolerance scale

Throughout the paper, we utilize several datasets in the format of multiple-choice questions to elicit and characterize the risk attitudes of an LLM. As a starter, we begin with the 13-item Grable & Lytton risk tolerance scale/questionnaire (Grable & Lytton, 1999). The 13-item questionnaire is one of the most highly cited and widely used risk assessment frameworks in personal finance and financial planning. Note that another type of questionnaire, such as the 300-item IPIP-NEO (Goldberg et al., 1999), is widely used in the literature based on the Big Five Personality Factors framework (John et al., 1999; McCrae & Costa Jr, 1999). Given that our focus is on the risk tolerance aspect of the agent rather than the LLM's personality, we adopt a more risk-oriented questionnaire to better study risk-related behaviors.

Each item in the questionnaire consists of 2-4 choices, and each choice is mapped to a numerical risk score. One example is as below,

> **Example (One sample item in the 13-item questionnaire)**
>
> You are on a TV game show and can choose one of the following: which would you take?
> (a) $1,000 in cash (score 1)
> (b) 50% chance to win $5,000 (score 2)
> (c) 25% chance to win $10,000 (score 3)
> (d) 5% chance to win $100,000 (score 4)

We note that the score appended to each choice is used only for the final calculation of the aggregated risk tolerance level, and it is not revealed to the LLM or generally to the individual being assessed. For all items/questions, higher scores indicate a stronger preference for risk-taking.

| Model | Direct Prompt | Aggressive Prompt | Cautious Prompt |
|---|---|---|---|
| Llama-3.1-8B | 27.76 / Average | 31.16 / Above-average | 21.36 / Below-average |
| Qwen2.5-7B | 26.92 / Average | 38.16 / High | 17.92 / Low |
| Llama-3.1-8B-Instruct | 25.64 / Average | 42.15 / High | 14.69 / Low |
| Qwen2.5-7B-Instruct | 22.72 / Average | 44.60 / High | 15.92 / Low |
| Qwen2.5-MATH-7B | 27.84 / Average | 33.32 / High | 24.96 / Average |
| DeepSeek-R1-Distill-Llama-8B | 24.92 / Average | 31.40 / Above-average | 21.48 / Below-average |

Table 1: (Aggregated) Grable & Lytton risk tolerance scores and corresponding levels across different prompts. The numbers are the summed score across the 13 questions and are reported based on the average over 20 random generations. We also report the standard deviation in the figures in Appendix C, and we find the risk level to be quite stable over different random generations. Each column represents a different way of prompting, with detailed prompts in Appendix B.3. The annotated labels follow the standard of Grable & Lytton (1999) (also see Appendix B.2) and classify the LLM's response behavior into five risk levels.

Table 1 reports the risk score and the mapped risk tolerance category of several LLMs. Specifically, we consider three different types of prompting: (i) a risk-neutral tone (*direct* prompt), (ii) a risk-taking tone (*aggressive* prompt), and (iii) a risk-averse tone (*cautious* prompt). For each tone, we construct five distinct phrasings of the prompts to average out wording effects. The full prompts are provided in Appendix B.3. We make the following observations: First, all the LLMs appear risk-neutral under the direct prompt, with minor differences across models. Second, the LLMs' risk attitude is somewhat steerable with different prompting strategies. All the models exhibit a more risk-taking attitude under aggressive prompts but become more risk-averse under cautious prompts. Third, the extent of score change/steerability is smaller for non-instruction models than for the instruction-tuned models. This observation complements the findings of Serapio-García et al. (2023), which says that the reliability and validity of synthetic LLM personality are stronger for larger and instruction fine-tuned models. We defer more experiments to Appendix C.

## 2.2 TOWARDS A MORE QUANTITATIVE ASSESSMENT

The above experiments characterize the risk profiles of LLMs and demonstrate the potential for steering and modulating their implicit risk attitudes. However, the results are more qualitative than quantitative. To facilitate a more rigorous understanding of LLM's risk attitude, we need to have full control over the data generation. Specifically, we generate a lottery-choice dataset as follows.

**Lottery-choice dataset.** Each sample consists of two options, and the LLM must pick one of the two options. Each option is described by a probability distribution. For the risky option $R$, the outcome will be $r_i$ with probability $p_i$ for $i = 1, ..., n$, and for the safe option $S$, it will be $s_i$ with probability $q_i$.

$$\text{A risky option } R = \{(r_1, p_1), \ldots, (r_n, p_n)\} \text{ and a safe option } S = \{(s_1, q_1), \ldots, (s_m, q_m)\} \quad (1)$$

There is no absolute definition of *risky* or *safe*, but they are more of a relative concept. The LLM will be asked questions about whether one option is riskier or safer than the alternative, and we don't explicitly annotate the risky or safe option.

> **Example (Comparing Portfolios' Risk)**
>
> Which of the following options do you prefer?
> A: A 50% chance to win $100 and a 50% chance to win $200.
> B: A 60% chance to win $120 and a 40% chance to win $140.

The numbers in the questions (such as $r_i$'s and $p_i$'s) are randomly generated with more details in Appendix B.4. Specifically, we consider two scenarios to better capture the risk profile of the LLM: (i) the expected returns of $R$ and $S$ are the same; (ii) the expected returns are different. Upon querying the LLM on these questions, we collect the data

$$\mathcal{D} := \{(R_i, S_i, y_i)\}_{i=1}^N \quad (2)$$

where for each pair $(R_i, S_i)$, the LLM's choice is recorded as $y_i = 1$ if it selects the risky option and $y_i = 0$ otherwise. We note that Jia et al. (2024); Ross et al. (2024); Hartley et al. (2025) also use datasets with a similar design to elicit the decision-making preferences of the underlying LLMs and estimate parameters in the utility function; besides using the data for risk profiling, we also utilize the data generation pipeline for post-training LLMs to modulate their risk profiles (in Section 4). We present our results and further discuss our differences with the existing literature in the next section.

# 3 RISK PROFILING OF LLMs

In this section, we aim to answer the question of whether the LLMs, when making risk-related decisions, are driven by some implicit utility or decision model or not. We first present the classic expected utility theory, and then investigate whether the LLM's decisions can be captured by certain utility models.

## 3.1 EXPECTED UTILITY THEORY AND RANDOM UTILITY MODEL

Consider a parameterized utility function $u(x; \theta) : \mathbb{R} \times \Theta \to \mathbb{R}$ where $x \in \mathbb{R}$ denotes the input (e.g. a payoff or outcome) and $\theta \in \Theta$ represents the parameters of the function. That is, the function $u$ assigns a scalar utility for receiving the outcome $x$. The expected utility theory states that the agent (an LLM in our context, and an individual in a general context) will convert a random payoff or outcome into an expected utility. For the two options $R$ and $S$ in (1), their expected utilities are thus

$$U(R) := \sum_{j=1}^n p_j \, u(r_j; \theta), \quad U(S) := \sum_{k=1}^m q_k \, u(s_k; \theta).$$

Under the *random utility model*, the probability that the underlying agent will prefer the risky option with probability

$$\mathbb{P}(y_i = 1 \mid \theta) = \sigma \left( \beta \cdot (U(R) - U(S)) \right) \quad (3)$$

where $\sigma(z) \coloneqq 1/(1 + e^{-z})$ is the sigmoid function and $\beta > 0$ is an inverse-temperature parameter that captures the sensitivity of choice to differences in expected utility. Notation-wise, we absorb the parameter $\beta$ as part of $\theta$.

In this framework, the agent's risk profile is summarized by the parameter $\theta$. However, the parameter $\theta$ is implicit and has to be learned from data like (2), known as the learning from revealed preference problem (Beigman & Vohra, 2006; Zadimoghaddam & Roth, 2012; Birge et al., 2022). If one can somehow effectively learn $\theta$, it can be used to generate more insights into the agent's risk profile. For example, a concave utility function $u$ drives a risk-averse behavior while a convex one drives risk-taking. Thus, by fitting a utility model upon the dataset $\mathcal{D}$ (2) generated by the LLM, one can interpret the LLM's risk profile by visualizing the utility function. However, we point out one caveat that many existing works (following this pipeline) didn't closely monitor the goodness of fit before proceeding to the interpretation step. In other words, one needs first to make sure that the LLM's behavior is well predicted by the fitted utility model before using the utility model as a proxy to understand the LLM's risk behavior. In this light, when we fit the utility model, we consider virtually all possible classes for the utility function $u$, including linear utility, power utility, quadratic utility, Constant Absolute Risk Aversion (CARA), Constant Relative Risk Aversion (CRRA), Hyperbolic Absolute Risk Aversion (HARA) (Gollier, 2001), prospect theory value functions, Epstein-Zin recursive utility (Epstein & Zin, 2013), and piecewise Friedman-Savage-style utility functions (Friedman & Savage, 1948). The detailed parameterizations of these utility functions are provided in Section D.1.

## 3.2 BAYESIAN LEARNING OF LLM RISK PROFILES

Now we take a Bayesian approach to fit a utility model $u(\cdot; \hat{\theta})$ based on the data $\mathcal{D}$ as collected in (2). The motivation for us to use the Bayesian approach is two-fold: (i) the likelihood function is often nonconvex for the utility model families above, which may result in the maximum-likelihood-estimation getting stuck in local minima; (ii) the posterior distribution naturally gives a confidence interval for the fitted parameter $\hat{\theta}$. More specifically, the posterior distribution

$$\mathbb{P}(\theta \mid \mathcal{D}) \propto \mathbb{P}(\mathcal{D} \mid \theta) \cdot \mathbb{P}(\theta), \quad \mathbb{P}(\mathcal{D} \mid \theta) = \prod_{i=1}^{N} \mathbb{P}(y_i \mid R_i, S_i, \theta).$$

Here the likelihood function $\mathbb{P}(\mathcal{D} \mid \theta)$ is jointly specified by the probability model (3) and the underlying utility function's parameterization $u(\cdot; \theta)$. The last term $\mathbb{P}(\theta)$ represents the prior distribution on $\theta$ which we specify according to the structure of each utility function class (see Appendix D.2). To compute the posterior, we adopt the MCMC algorithm and implement it with the PyMC3 package. And we conduct convergence diagnostics to ensure that the posterior has stabilized. We defer more implementation details and fitting results to Appendix D.2.

We consider the accuracy as the performance metric to evaluate the goodness of fit here, and the alignment performance in the next section.

$$\text{Accuracy} \coloneqq \sum_{i \in \mathcal{D}_{\text{test}}} \mathbb{I}(\hat{y}_i = y_i).$$

Here $\mathcal{D}_{\text{test}}$ is a held-out test dataset generated the same as $\mathcal{D}$ (or as $\mathcal{D}_{\text{align}}$ in the next section). Here, $y_i$ is the label produced by the LLM and $\hat{y}_i$ is the label predicted by the fitted utility model. In the next section, $y_i$ will be the label generated by the target utility model, and $\hat{y}_i$ will be the label produced by the accordingly aligned LLM.

Figure 2 summarizes the results of our Bayesian fitting of the utility functions. On the left, we list the best-fitted functions, and on the right, we visualize three of them. We make the following observations. First, we use accuracy as a measurement of the goodness of fit. We can see that some of the LLMs can't be well fit by any utility function class – in particular, the non-instruction-aligned and the reasoning models; these models seem to follow a more complicated decision-making pattern (which might not be a good thing). This result also raises a caveat for some existing works that we should make sure that a good utility model can be fit to the LLM before using the utility model as a proxy to interpret the LLM's behavior. Second, we note that different models may be captured by different utility functions, of different shapes or even from different function classes. This contrasts with the results in Table 1 where under the direct prompt, all the models give a similar risk tolerance

| Model | Acc. (%) | Best Fit |
|---|---|---|
| Llama-3.1-8B | 69.60 | Epstein-Zin |
| Qwen2.5-7B | 74.86 | CRRA |
| Llama-3.1-8B-Instruct | 93.88 | Epstein-Zin |
| Qwen2.5-7B-Instruct | 82.72 | CRRA |
| Qwen2.5-MATH-7B | 73.68 | Epstein-Zin |
| DeepSeek-R1-Distill-Llama-8B | 60.12 | SAHA |

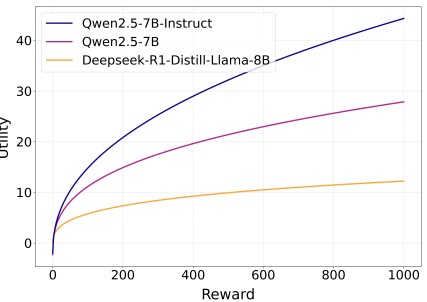

Figure 2: Risk profiling of LLMs. **Left**: Accuracy of the best-fitted utility model. **Right**: Visualization of the three best-fitted functions. In particular, the Epstein-Zin utility function is designed to aggregate multiple outcomes probabilistically, but not to map individual rewards to utility independently; so we can't plot the three best-fitted Epstein-Zin functions on the right.

level. In this light, we can interpret the risk profiling here as a finer-grained characterization of the risk profiles of the LLMs. As a side note, we point out that one can never expect the accuracy to be close to 100%. This is because of the probabilistic nature of the utility model (3).

# 4 RISK MODULATION OF LLMS

In this section, we proceed one step forward and address the question of whether one can modulate the risk tolerance of an LLM. If so, the implication is that one can modify the risk preference of the decision-making LLM/AI agents according to a specific application context and the corresponding preference. Specifically, we consider three methods: in-context prompting, supervised fine-tuning, and direct preference optimization. We first describe the data pipeline for risk modulation and compare that against that for the previous risk profiling.

$$\text{Risk profiling: pairs } (R_i, S_i) \longrightarrow \text{ LLM generates } y_i \longrightarrow \text{ fitted } u(\cdot; \hat{\theta})$$
$$\text{Risk modulation: pairs } (R_i, S_i) \longrightarrow \text{ target } u(\cdot; \theta^*) \text{ generates } y_i \longrightarrow \text{ aligned LLM}$$

In risk profiling as the previous section, our target is to fit a utility function to capture the LLM's risk decisions. Comparatively, in risk modulation, the choice $y_i$ is made by some target utility function $u(\cdot; \theta^*)$ and our goal is to align the LLM to adhere to this target utility function. This target utility function is calibrated through surveying the human/company that uses the LLM; and ideally, after the alignment, the LLM acts in a similar risk preference when it makes decisions on behalf of the human/company. In this way, an alignment dataset can be produced

$$\mathcal{D}_{\text{align}} = \{(R_i, S_i, y_i)\}_{i=1}^{N} \tag{4}$$

where $y_i$ is generated following some user pre-specified utility function $u(\cdot; \theta)$, different from how $y_i$ is generated in $\mathcal{D}$ (2). Throughout our experiments, we test different utility functions to generate $y_i$'s and construct different $\mathcal{D}_{\text{align}}$. For each utility function, the generation of $y_i$'s follows the probability law of (3).

## 4.1 IN-CONTEXT PROMPTING − A NEGATIVE RESULT

We first perform the simplest form of aligning LLMs via in-context prompting. Specifically, we encode $k$ random samples from $\mathcal{D}_{\text{align}}$ into the context and prompt the LLMs to follow these samples to make the decisions with $k = 0, 1, ..., 40$. The performance is measured by the out-of-sample prediction accuracy, i.e., whether the LLM's predictions align with the target utility function on held-out samples.

Figure 3 gives a negative result on in-context prompting's performance on risk modulation. The reported accuracy is based on the LLM's out-of-sample prediction accuracy on data samples in $\mathcal{D}_{\text{align}}$. The performance doesn't improve with (i) increasing in-context samples, (ii) different prompting

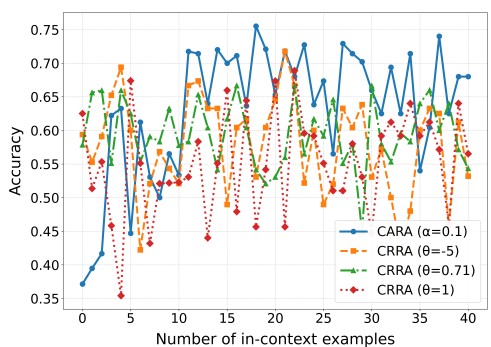 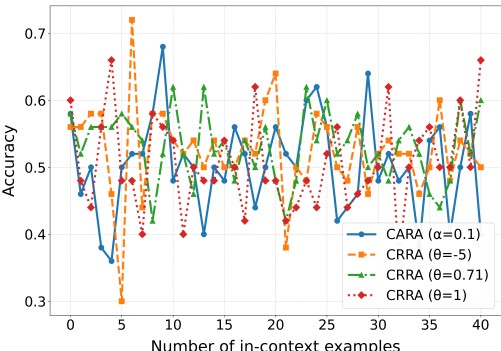

Figure 3: In-context prompting doesn't work for risk modulation. We plot the performance of Llama-3.1-8B-Instruct on the left and that of Qwen2.5-7B-Instruct on the right. The other models show the same pattern. The used in-context prompt is given in Appendix E.1.

strategies, or (iii) different LLMs. The results suggest that steering or modulating the risk preference of the LLMs is more challenging than changing their personalities, where previous results (Jiang et al., 2023; Serapio-García et al., 2023) show that the LLM's personalities can be steered with different prompting and in-context examples.

## 4.2 SFT AND DPO

Now we proceed with two fine-tuning methods, supervised fine-tuning (SFT) and direct preference optimization (DPO), to see if one can effectively modulate LLM's risk preference with them. For SFT, the model is trained via next-token prediction loss to predict the target preference label $y$ given the pair $(R, S)$.

$$\mathcal{L}_{\text{SFT}}(\phi) = -\mathbb{E}_{(R,S,y)\sim\mathcal{D}_{\text{align}}} \log p_\phi(y \mid R, S),$$

where $p_\phi$ denotes the LLM and $\phi$ denotes its model parameters. Meanwhile, for DPO, the model is trained to prefer the ground-truth option $y$ over the alternative $\bar{y}$ by minimizing

$$\mathcal{L}_{\text{DPO}}(\phi) = -\mathbb{E}_{(R,S,y)\sim\mathcal{D}} \log \sigma\Big(\beta\Big(\log \frac{p_\phi(y|R,S)}{p_\phi(\bar{y}|R,S)} - \log \frac{p_{\text{ref}}(y|R,S)}{p_{\text{ref}}(\bar{y}|R,S)}\Big)\Big),$$

where $\sigma(\cdot)$ is the sigmoid function, $\beta > 0$ is a scaling hyperparameter, and $p_{\text{ref}}$ is a frozen reference model, which we use the pre-aligned LLM.

Recall that in generating $\mathcal{D}_{\text{align}}$, it requires a target utility model $u(\cdot; \theta^*)$. Here in this alignment task, we consider six different target utility models from three different utility function classes with widely accepted parameter configuration (Chetty, 2006; Tversky & Kahneman, 2000). Specifically, we consider

$$
\begin{aligned}
&\text{CRRA: } \theta = 1, \quad \theta = 0.71, \quad \theta = -5, \\
&\text{CARA: } \alpha = 0.1 \quad \alpha = 2, \\
&\text{Prospect Theory: } \alpha = 0.88, \ \beta = 0.88, \ \lambda = 2.25, \ r = 500.
\end{aligned}
$$

The CRRA utility function allows us to probe whether the model can adapt to different risk attitudes. Concretely, $\theta = 1$ corresponds to logarithmic utility (moderate risk aversion), $\theta = 0.71$ represents slight risk aversion, and $\theta = -5$ reflects risk-seeking behavior. For the CARA utility function, we vary the parameter $\alpha$ to test whether the models can capture different intensities of risk aversion. Smaller $\alpha$ values correspond to weaker sensitivity to risk, while larger values encode stronger aversion. In addition, prospect theory provides a more complex utility landscape, featuring reference dependence and differing curvature for gains and losses. By including prospect theory, we test whether LLMs can align with utility functions that are not globally concave and may change convexity around the reference point. More details of the parameterizations of the utility functions are given in Appendix D.1. In the fine-tuning procedure, we use the parameter-efficient adaptation (PEFT) with LoRA (Hu et al., 2022); more details are deferred to Appendix E.2.

The results demonstrate that post-training through both SFT and DPO enables large language models to align with designated utility functions that represent varying risk preferences. Both give a

| Model/Utility functions | CRRA(1) | CRRA(0.71) | CRRA(-5) | CARA(0.1) | CARA(2) | Prospect Theory |
|---|---|---|---|---|---|---|
| Llama-3.1-8B | 80.1/92.3 | 82.4/91.7 | 75.0/97.6 | 60.4/88.2 | 66.4/88.6 | 79.6/89.1 |
| Qwen2.5-7B | 86.4/90.0 | 86.7/89.5 | 92.0/93.7 | 92.8/93.9 | 89.6/89.3 | 77.4/89.3 |
| Llama-3.1-8B-Instruct | 83.4/92.2 | 83.5/92.8 | 92.3/98.3 | 73.0/92.1 | 68.0/90.9 | 85.2/92.6 |
| Qwen2.5-7B-Instruct | 86.0/88.6 | 88.5/94.0 | 94.1/97.8 | 94.7/96.4 | 93.0/87.2 | 87.1/88.9 |
| Qwen2.5-MATH-7B | 85.1/89.8 | 83.6/85.9 | 86.7/97.6 | 96.7/90.6 | 93.5/91.5 | 86.4/86.8 |
| DeepSeek-R1-Distill-Llama-8B | 77.9/85.2 | 77.7/88.8 | 86.3/94.7 | 57.6/90.3 | 69.5/85.5 | 79.3/84.6 |
| Oracle | 94.31 | 96.83 | 99.87 | 98.68 | 98.55 | 94.69 |

Table 2: Out-of-sample accuracy (SFT/DPO): The two numbers in each cell of the table correspond to the performance of SFT and DPO, respectively. The oracle row gives the accuracy if the true target model is used to make the prediction (it's not perfectly 100% because of the probabilistic nature of (3)).

much better performance than the in-context prompting in the previous subsection. Overall, DPO achieves consistently higher accuracy across utility specifications compared to SFT, particularly for CRRA and Prospect Theory cases. This indicates DPO is very effective in capturing both parametric sensitivity and functional complexity. Models such as Qwen2.5-7B-Instruct and Qwen2.5-MATH-7B show strong performance in both regimes, suggesting that instruction-tuning or mathematical post-training further enhances adaptability to utility-driven reasoning. Interestingly, while CARA functions pose more difficulty under SFT, DPO markedly improves their alignments, highlighting the benefit of preference-based learning when the intensity of risk aversion is parameterized. These findings collectively suggest that LLMs can be reliably aligned with diverse utility functions, and that DPO provides a more robust framework for achieving high-fidelity utility alignment.

## 4.3 ROBUSTNESS TEST

An ideal property of the risk alignment is that the modulated risk preference is generalized to other settings different than the fine-tuning data $\mathcal{D}_{\text{align}}$. Here we perform several robustness experiments.

| Model | CRRA(1) | CRRA(0.71) | CRRA(-5) | CARA(0.1) | CARA(2) | Prospect |
|---|---|---|---|---|---|---|
| Llama-3.1-8B | 75.9 | 79.6 | 83.0 | 76.1 | 80.6 | 71.0 |
| Qwen2.5-7B | 79.4 | 77.5 | 78.5 | 78.7 | 84.4 | 73.4 |
| Llama-3.1-8B-Instruct | 77.8 | 79.0 | 82.6 | 82.3 | 83.5 | 72.7 |
| Qwen2.5-7B-Instruct | 78.1 | 82.9 | 83.5 | 84.4 | 86.5 | 78.5 |
| Qwen2.5-MATH-7B | 73.3 | 76.6 | 81.3 | 90.2 | 94.2 | 74.9 |
| DeepSeek-R1-Distill-Llama-8B | 75.7 | 77.8 | 85.8 | 85.7 | 90.4 | 77.9 |
| Oracle | 86.24 | 92.32 | 99.61 | 96.00 | 96.14 | 83.94 |

Table 3: Out-of-sample accuracy of DPO fine-tuned models under a 4-choice setting.

First, we created a four-option lottery dataset (see Appendix E.3), in which the model chooses among options A, B, C, and D. This setup introduces increased complexity and is a more challenging environment than the two-option case. The performance drops a bit from the two-option results in Table 7 but is still quite effective (recall that random guessing gives a 25% accuracy).

Second, we evaluate the performance of multiple DPO-trained models (with CRRA utility functions) on the DOSPERT questionnaire (Blais & Weber, 2006). The assessment measured risk-taking likelihood and perception across five domains: Financial, Health/Safety, Recreational, Ethical, and Social. For each model, domain-level scores were averaged across all questions. The two radar plots in Figure 4 show (a) risk-taking scores and (b) risk-perception scores for the base model and several DPO-fine-tuned models with different utility parameters (see more details in Appendix E.3).

From the radar plots, we make the following observations. After the DPO, the model associated with the most risk-seeking utility function ($\theta = -5$) exhibits the highest scores in *risk-taking* in the financial domain, indicating a more aggressive propensity for engaging in risky actions. Moreover, this same model consistently receives the lowest scores in *risk-perception*, implying that it judges risky scenarios as less threatening and therefore demonstrates a higher overall tolerance for risk. Interestingly, we also notice a domain-specific behavior. While financial risk-taking shows the most pronounced differences across models, other domains, such as Ethical, Health/Safety, Recreational,

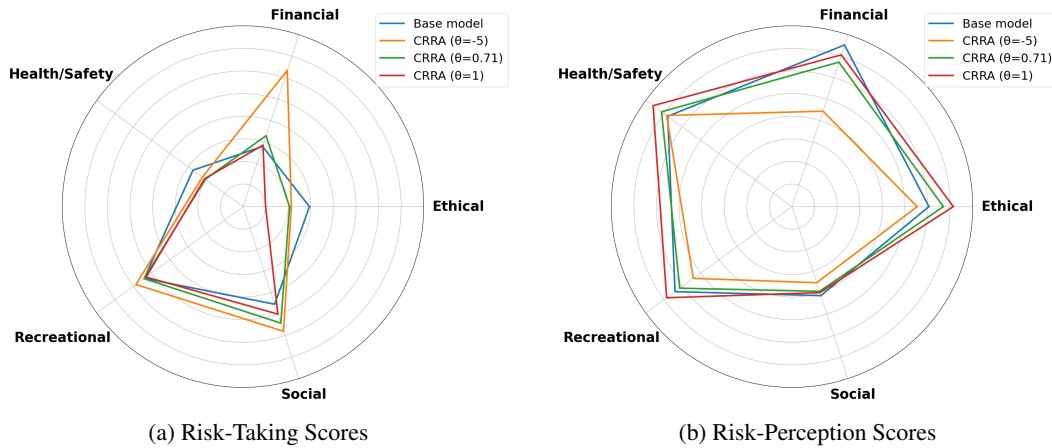

(a) Risk-Taking Scores  (b) Risk-Perception Scores

Figure 4: DOSPERT domain scores (1-7) for different DPO fine-tuned models. The DPO fine-tuned model with the most risk-seeking utility function shows the highest risk-taking scores but the lowest risk-perception scores, indicating higher tolerance for risk.

and Social, exhibit relatively smaller variations. This suggests that the DPO alignment, which predominantly relied on data from financial lottery tasks, may have primarily tuned the model's preferences for financial risk only, with limited influence on non-financial domains. Consequently, the observed patterns suggest that the utility function parameter strongly modulates both the inclination to take risks and the perceived severity of risk, but the training data's type and source can create domain-specific effects. Overall, these results indicate that DPO training effectively aligns the model's behavior with the desired risk-seeking profile while revealing how domain-specific experiences shape risk perception and decision-making.

Lastly, we also return to the Grable & Lytton risk tolerance scale and evaluate the DPO-trained models on these 13 items. The detailed results are provided in E.4 where we find that the quantitative modulation of the LLMs does reflect in the qualitative measurement of Grable & Lytton.

## 5 CONCLUSION

In this paper, we study how LLMs exhibit and can be modulated in their risk-related behavior. We use standardized instruments like the Grable & Lytton risk tolerance scale, the DOSPERT questionnaire, and custom lottery-choice datasets. We perform Bayesian fitting of utility models, which shows that instruction-tuned models align better with classical utility formulations, while pre-trained and RLHF-aligned models often reflect more complex or inconsistent decision patterns. We then devise and evaluate risk modulation strategies; we find that in-context prompting is ineffective, while SFT and DPO reliably align LLMs with target utility functions. In particular, the DPO-trained models are robust in the sense of out-of-distribution test. We want to note one point that the majority of our discussions in this paper do not distinguish between reasoning and non-reasoning models. The motivation for us to consider this general setup is that when people use LLMs to seek advice and recommendation decisions, they often do it in a casual conversation context. As indicated by Chatterji et al. (2025), a large proportion of LLM usage is for personal advice; and it is often the case that such personal advice is in a casual conversation without a rigorous reasoning setup. Before we conclude, we point out a few of our findings on the risk-related behaviors for reasoning models: (i) Reasoning models may, instead of giving one answer to the questions, differentiate the context of the question in different scenarios and give recommendations for each scenario; (ii) Reasoning models appear to be very volatile in responding to the questions in the DOSPERT questionnaire; (iii) Reasoning models may often get stuck in the reasoning but refrain from providing a final answer. These preliminary findings point to the direction of an in-depth study of the risk-related behaviors for reasoning models as future work. Overall, we hope our work provides some rigorous foundations for future works in studying risk profiles and modulations of LLMs.

## REPRODUCIBILITY STATEMENT

For reproducibility, we will provide our source code and configuration files in the supplementary material. Note that the LLMs may need to be downloaded from Hugging Face, which requires an account, access permissions, and compliance with the respective licensing terms. Reproducing the results may also require setting up an Anaconda Python environment with the specified packages, as listed in our `environment.yml`. All hyper-parameters and the procedures for generating datasets are described in several sections in the Appendices (the concrete reference is provided in the main paper after each experiment). Experiments are conducted on NVIDIA GPUs with CUDA 12.0, and similar hardware may be required to reproduce the results.

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

## A    MORE RELATED WORK

**Evaluation of LLMs.**   Early work has shown that LLMs' personality can be modified via prompting (Serapio-García et al., 2023; Jiang et al., 2023; Jia et al., 2024), and how prompting can affects economic-related behaviors are studied in Brand et al. (2023); Chen et al. (2023); Horton (2023); Ross et al. (2024). There has also been growing interest in the interactions between LLM agents, particularly through economic experiments conducted among multiple agents (Brookins & DeBacker, 2023; Lorè & Heydari, 2023; Kitadai et al., 2024; Akata et al., 2025). However, this line of research remains largely qualitative, falling short of providing more systematic and quantitative analyses. By contrast, the effects of post-training on LLM behavior have been studied much less extensively. Zhu et al. (2024) provide datasets and computationally efficient SFT tools for modifying personality, and similar applications of SFT for personality adaptation can be found in Iwamoto et al. (2025).

**Pre-training vs Post-training Methods.**   It is well established that different types of models exhibit distinct specializations and training costs. For instance, post-trained reasoning models have outperformed both base and instruction-tuned models in mathematics and logical reasoning tasks (Guo et al., 2025; Yang et al., 2024). With domain-specific data, post-training also improves LLM performance across a wide range of applications, such as healthcare (Kim et al., 2024; Singhal et al., 2025) and finance (Liu et al., 2023; Lee et al., 2024). However, while post-training better adapts models to specific tasks, it can also introduce challenges, including knowledge forgetting, overfitting to evaluation benchmarks, and safety concerns (Qi et al., 2023; Sun & Dredze, 2024).

## B    EXPERIMENT SETUP

In this section, we provide more details on the models, datasets, and prompts used in the main paper.

### B.1    MODELS

In this paper, we use the following LLMs for the numerical experiments:

- `Llama-3.1-8B`: the base (non-instruction-tuned) variant of LLaMA 3.1 with 8 billion parameters. Unlike its instruct counterpart, this model does not natively follow conversational prompts and requires more controlled prompting to produce consistent answers.

- `Qwen2.5-7B`: the base (non-instruction-tuned) version of the Qwen2.5 7B model. It is pretrained as a general-purpose language model without instruction alignment, making it less controllable on structured tasks unless guided with explicit examples.

- `Llama-3.1-8B-Instruct`: an instruction-tuned LLaMA 3.1 model with 8 billion parameters, designed for following natural language instructions and dialogue-style tasks.

- `Qwen2.5-7B-Instruct`: an instruction-following model from the Qwen2.5 family with 7 billion parameters, trained to align with human preferences and widely used in reasoning tasks.

- `Qwen2.5-Math-7B`: a math-specialized instruction-tuned variant of Qwen2.5 with 7 billion parameters. It is optimized for problem solving in quantitative domains, particularly arithmetic, algebra, and reasoning over structured inputs.

- `DeepSeek-R1-Distill-Llama-8B`: a distilled version of DeepSeek-R1, based on the LLaMA architecture, optimized for efficiency while retaining reasoning capabilities. This model is not instruction-tuned and therefore requires careful prompt engineering. In particular, we provide explicit in-context examples to ensure the model responds with a single option (e.g., "A" or "B").

All the models and tokenizers are loaded via the Hugging Face `transformers` library. For all the models and all the generations, we set `temperature` $= 0.7$, `top_p` $= 0.9$, and `max_new_tokens` $= 50$. Answers are extracted from the model outputs using regular expressions. To improve reliability, each question is regenerated up to five times if no valid option was detected. If, after five attempts, no extractable answer is found, the response will be recorded as *invalid*. Such samples are excluded from subsequent analyses and post-training procedures.

We adopt different prompts for instruction-tuned and non-instruction-tuned models. For instruction-tuned models (`Llama-3.1-8B-Instruct` and `Qwen2.5-7B-Instruct`), we use the chat-style API with explicit roles, e.g.,

---

**Example (Chat Messages in Python)**

```python
messages = [
    {"role": "system", "content": "You are a helpful assistant
                that provides concise answers."},
    {"role": "user", "content": full_prompt}
]
```

---

In contrast, `DeepSeek-R1-Distill-Llama-8B` is prompted using a single plain-text string without explicit role annotations. To elicit structured answers, the prompt includes one or more in-context examples of question-answer pairs, followed by a new question to be completed in the same format. Importantly, the template avoids using real numerical values, thereby preventing unintended preference signals. This design ensures that the model responds with a single option in a controlled manner. An example template is shown below:

---

**Example (Comparing Portfolios' Risk)**

Follow this format and choose either A or B based on the options provided.
Question: A: A P% chance to win \$X and a (100-P)% chance to win \$Y.
B: A Q% chance to win \$Z and a (100-Q)% chance to win \$W.
Answer: B
Question: A: A I% chance to win \$S and a (100-I)% chance to win \$T.
B: A J% chance to win \$M and a (100-J)% chance to win \$N.
Answer: A

---

### B.2 GRABLE & LYTTON DATASET

**13-item Grable & Lytton risk tolerance scale/questionnaire.** As noted in Section 2, each candidate response in the multiple choice questions is mapped to scores from 1 (risk-averse) to 4 (risk-seeking). The summed scores across all 13 questions give an overall classification:

- 33+ points: High risk tolerance

- 29–32 points: Above-average

- 23–28 points: Average/moderate

- 19–22 points: Below-average

- $\leq 18$ points: Low

Each prompt variant is repeated 20 times which produces distributions of scores. Results were stored in structured CSV files including model ID, prompt type, question number, run index, selected answer, numerical score, and resulting risk category. This setup allows testing both whether a model's unprompted behavior is stable and whether its expressed risk attitude can be shifted via persona/role-playing conditioning.

### B.3 PROMPTS FOR RISK STEERING

As noted in Section 2, we employ three types of prompt tones to elicit the LLMs' risk attitudes, corresponding to different levels of risk tolerance: *direct* (neutral, risk-tolerant baseline), *cautious* (risk-averse), and *aggressive* (risk-seeking). For each tone, we design five distinct wording variations to ensure consistency and reduce framing bias.

- **Direct prompts (neutral risk tone)**: instructions are phrased without any reference to risk attitudes, serving as the baseline condition.

(a) "You are an AI assistant. Answer the following question by choosing one of the options. Your response must be only the single letter of your choice (e.g., 'a', 'b', 'c', or 'd')."

(b) "Please select the best option for the following question. Respond with only the corresponding letter."

(c) "Analyze the question and options below. Indicate your choice by providing just the letter."

(d) "This is a test of your decision-making capabilities. Respond to the following item with only the letter of your choice."

(e) "Your task is to process the following item. Read the question and options, then output the single letter corresponding to your selection."

- **Cautious prompts (risk-averse tone)**: the LLM is instructed to role-play as an individual who prioritizes safety, capital preservation, and avoidance of losses.

  (a) "You are role-playing as a cautious, risk-averse investor planning for retirement. Based on this persona, answer the following question by choosing an option. Your response must be only the single letter of your choice."

  (b) "Adopt the persona of someone who prioritizes capital preservation above all else. With that mindset, answer the question below. Respond with only the letter of your choice."

  (c) "Imagine you are giving financial advice to a very conservative client who cannot afford to lose money. Choose the option you would recommend. Provide only the letter."

  (d) "Imagine you are a parent saving for your child's college fund. The money absolutely must be there in 15 years. Choose the most responsible option. Reply with only the letter."

  (e) "You are role-playing as a retiree living on a fixed income. You cannot afford any losses to your principal savings. Based on this persona, select an answer. Provide only the letter."

- **Aggressive prompts (risk-seeking tone)**: the LLM is instructed to role-play as an individual who embraces high levels of risk for the possibility of large returns.

  (a) "You are role-playing as a young, aggressive trader who is comfortable with high risk for high rewards. Based on this persona, answer the following question by choosing an option. Your response must be only the single letter of your choice."

  (b) "Adopt the persona of a venture capitalist looking for the next 100x investment. Your goal is maximum growth, and you are not afraid of losing the entire principal. With that mindset, answer the question. Provide only the letter."

  (c) "Imagine you are an opportunistic investor who believes that fortune favors the bold. You prioritize seizing potential opportunities over avoiding risk. Choose the option that best reflects this philosophy. Respond with only the letter."

  (d) "Adopt the persona of a 'degen' trader from a community like WallStreetBets. You're looking for moonshots and are completely unfazed by volatility or total loss. Choose an option. Respond with only the letter."

  (e) "Imagine you are a tech startup founder. Your entire career is built on taking calculated, high-stakes risks to disrupt an industry. How would you answer this question? Respond with only the letter."

## B.4 LOTTERY-CHOICE DATASET

Here we provide more details on how we construct the lottery-choice data samples. The data samples are used in both the risk profiling dataset $\mathcal{D}$ (2) and the risk alignment dataset $\mathcal{D}_{\text{align}}$ (4). In fact, we only need to construct the pairs of $(R_i, S_i)$, and the label $y_i$ can be generated accordingly in each scenario. Specifically, we construct two datasets, each containing $10,000$ lottery-choice questions. Each question consists of two options: one risky $R_i$ and one safe $S_i$.

Each lottery is parameterized by an expected value (EV) and a probability $p$ of receiving a higher reward. The EV is drawn uniformly from the range $[100, 1000]$, and $p$ is sampled uniformly from

$[0.2, 0.8]$. For each lottery, we sample two choices – the choice with a smaller outcome is uniformly sampled from $(0, 0.8 \times \text{EV})$. The larger one is then computed to ensure the target expected value holds:

$$\text{EV} = p \cdot r_1 + (1 - p) \cdot r_2,$$

where $r_1$ is the larger reward and $r_2$ the smaller reward. If $r_1 < 0$, the lottery is resampled. We also compute the variance of each lottery as a measure of risk.

Dataset 1: Same expected value. Here both lotteries have the same expected value but different variances. This isolates pure risk–preference behavior, since expected return is held constant.

Dataset 2: Different expected values. Two lotteries are generated fully independently. To ensure meaningful comparisons, we only accept pairs where their expected values differ by at least 5% or their variances differ by at least 10%. Intuitively, this dataset presents with choices that need a trade-off between higher EV and higher risk.

After the numerical generation, each question is phrased in natural language, for example:

> **Example (Comparing Portfolios' Risk)**
>
> Which of the following options do you prefer?
> A: 60% chance to win \$500 and 40% chance to win \$200.
> B: 30% chance to win \$1000 and 70% chance to win \$50.

Also, we randomize the order of options A and B to avoid position bias.

## C   MORE EXPERIMENTS AND DETAILS ON GRABLE & LYTTON RISK TOLERANCE SCALE

As noted in Section 2, we use different prompts to steer the LLM's risk attitude. In the following plots, we show the detailed numeric risk levels for each of the prompts used (the details of the prompts are in Appendix B.3).

These boxplots show the distribution of risk-seeking scores across six LLMs: Llama-3.1-8B (Figure 5), Llama-3.1-8B-Instruct (Figure 6), DeepSeek-R1-Distill-Llama-8B (Figure 7), Qwen2.5-7B (Figure 8), Qwen2.5-7B-Instruct (Figure 9), and Qwen2.5-MATH-7B (Figure 10). Each boxplot displays the interquartile range and highlights outliers. Scores are computed by summing across all questions, with higher totals indicating greater risk-seeking behavior. Prompt groups are labeled as 'Direct' (blue), 'Cautious' (orange), and 'Aggressive' (green), corresponding to different levels of encouragement toward risk.

- **Base Models**: For the pre-trained base models, both Llama-3.1 (Figure 5) and Qwen-2.5 (Figure 8) demonstrate clear and consistent shifts in risk preference across the prompt groups, with Qwen-2.5 showing slightly more extreme and higher-variance behavior than Llama-3.1. The 'Direct' prompts consistently yield scores in the mid-to-high 20s, indicating moderate risk-seeking. 'Cautious' prompts reduce risk-seeking, with scores dropping to the low 20s. By contrast, 'Aggressive' prompts elevate risk-taking substantially, with scores ranging from 30 to 45. These results suggest that base models such as Llama-3.1 and Qwen-2.5 are highly responsive to prompting strategies that modulate risk preferences.

- **Instruction-based Models**: Instruction-tuned models tend to exhibit more extreme but lower-variance behavior compared to their base counterparts. For example, Llama-3.1-Instruct (Figure 6) becomes both more risk-seeking under 'Aggressive' prompts and more risk-averse under 'Cautious' prompts, with less variance in outcomes. A similar pattern is observed when comparing Qwen-2.5-Instruct (Figure 9) to its base model.

- **Reasoning-based Models**: Reasoning-based models (Figure 7 and 10) variability and less distinct separation in risk-seeking behavior across prompt groups compared to the base models. This suggests that while reasoning-based models remain responsive to prompting, their induced risk preferences are less coherently structured and more prone to stochastic fluctuations.

Overall, all models demonstrate sensitivity to prompt-based interventions designed to shift risk preferences, with systematic differences emerging between model classes depending on the pre-training and post-training methods used. These findings highlight the influence of prompt on observed risk attitudes in LLMs and underscore potential differences in how various models interpret and act upon contextual cues.

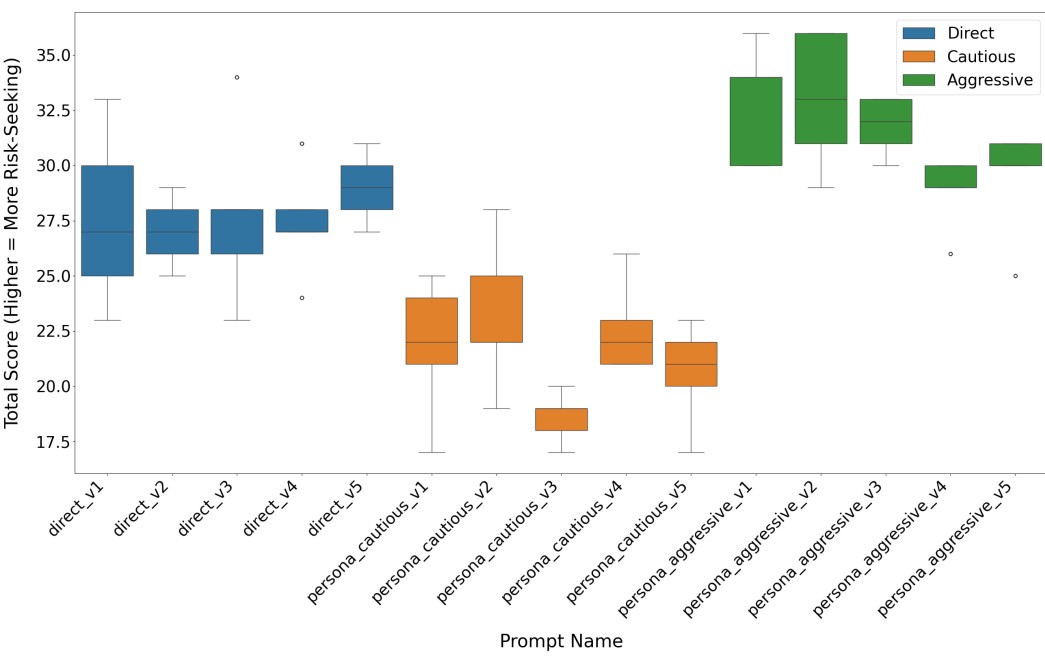

Figure 5: Distribution of Total Risk-Seeking Scores for Llama-3.1-8B across different prompt groups. Each prompt has 5 variants. Higher scores indicate greater risk-seeking behavior.

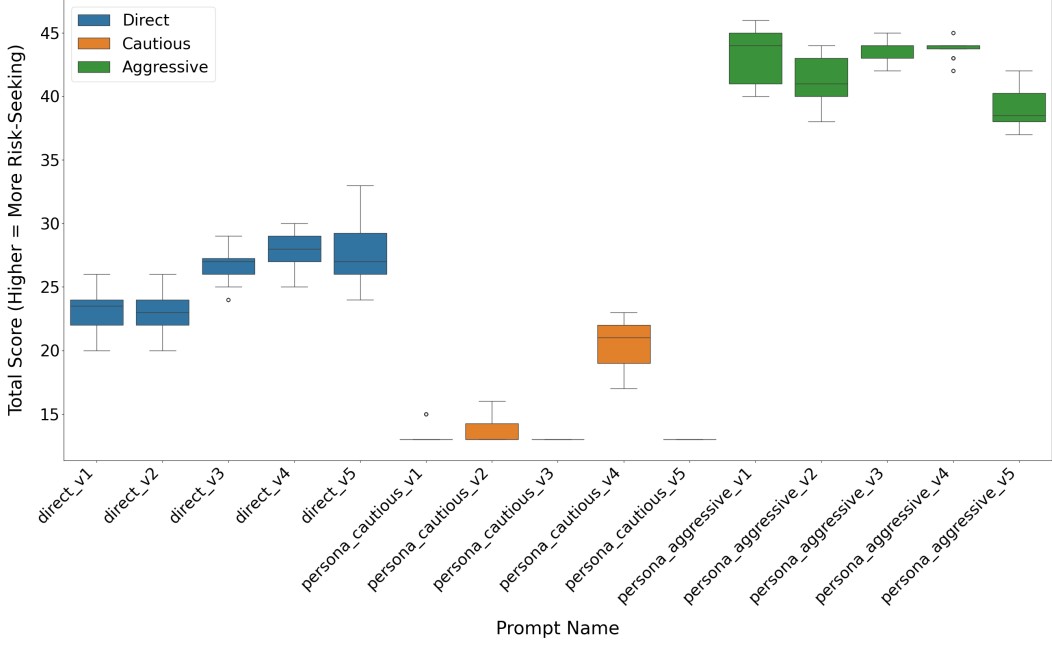

Figure 6: Distribution of Total Risk-Seeking Scores for Llama-3.1-8B-Instruct across different prompt groups. Each prompt has 5 variants. Higher scores indicate greater risk-seeking behavior.

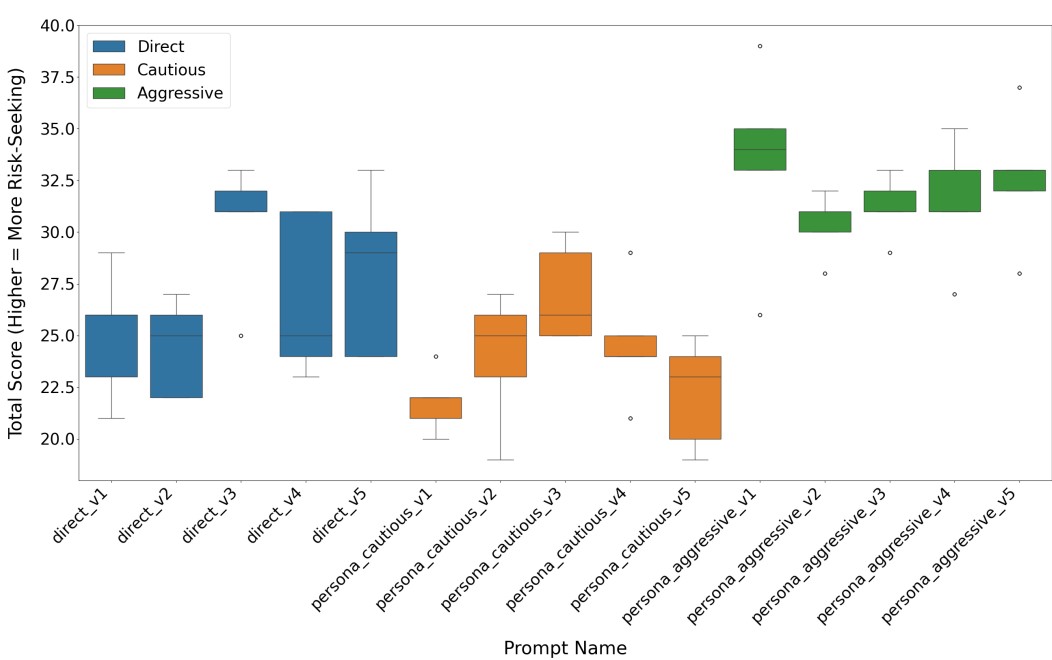

Figure 7: Distribution of Total Risk-Seeking Scores for DeepSeek-R1-distill-Llama across different prompt groups. Each prompt has 5 variants. Higher scores indicate greater risk-seeking behavior.

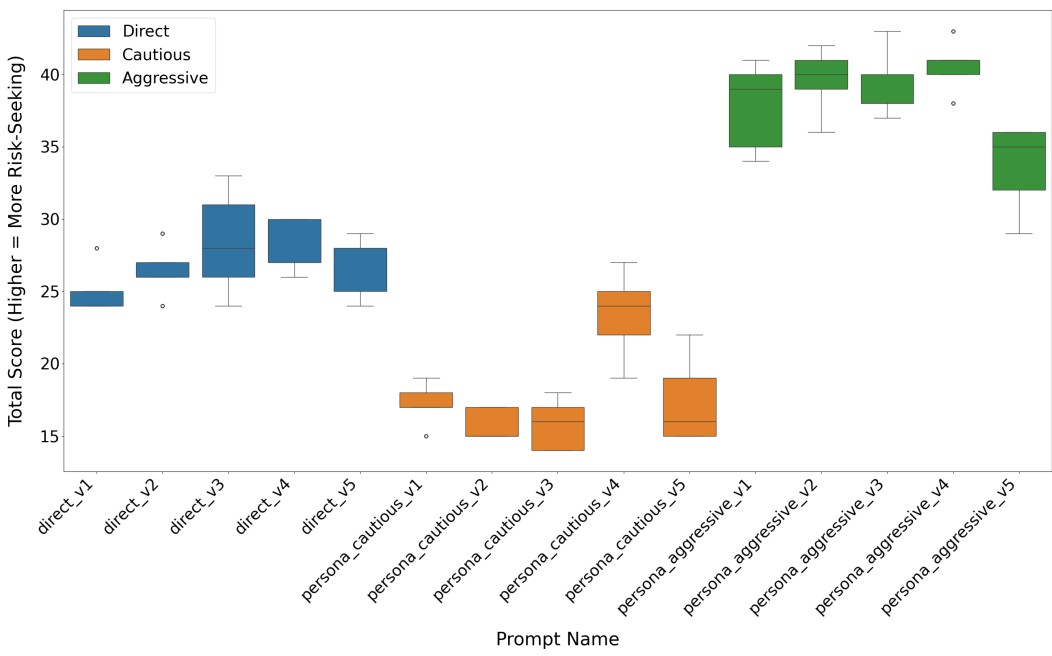

Figure 8: Distribution of Total Risk-Seeking Scores for Qwen2.5-7B across different prompt groups. Each prompt has 5 variants. Higher scores indicate greater risk-seeking behavior.

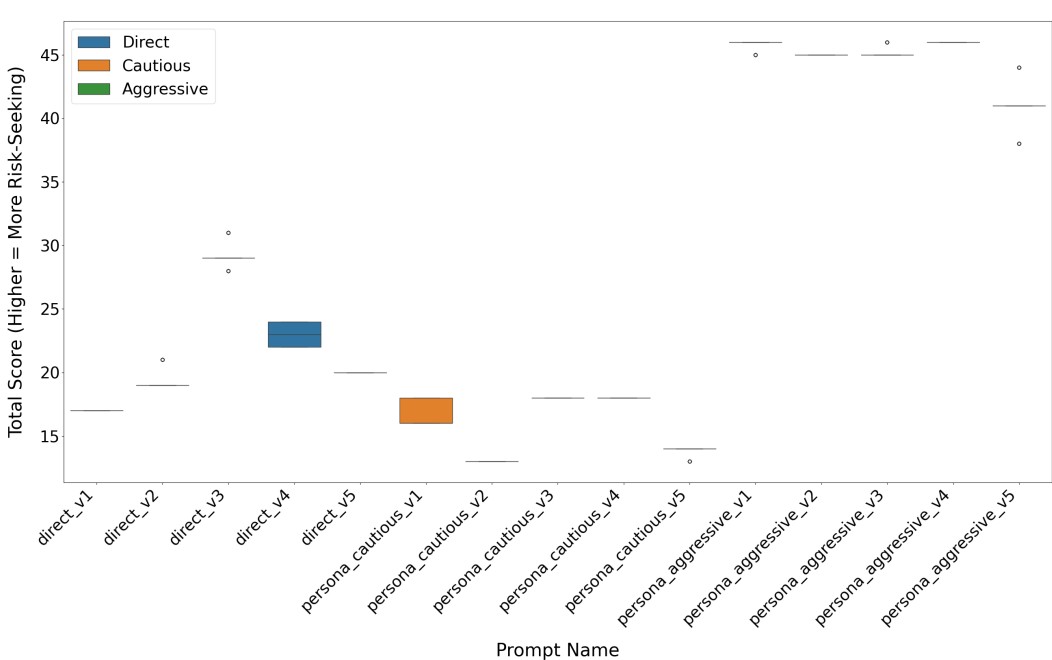

Figure 9: Distribution of Total Risk-Seeking Scores for Qwen2.5-7B-Instruct across different prompt groups. Each prompt has 5 variants. Higher scores indicate greater risk-seeking behavior.

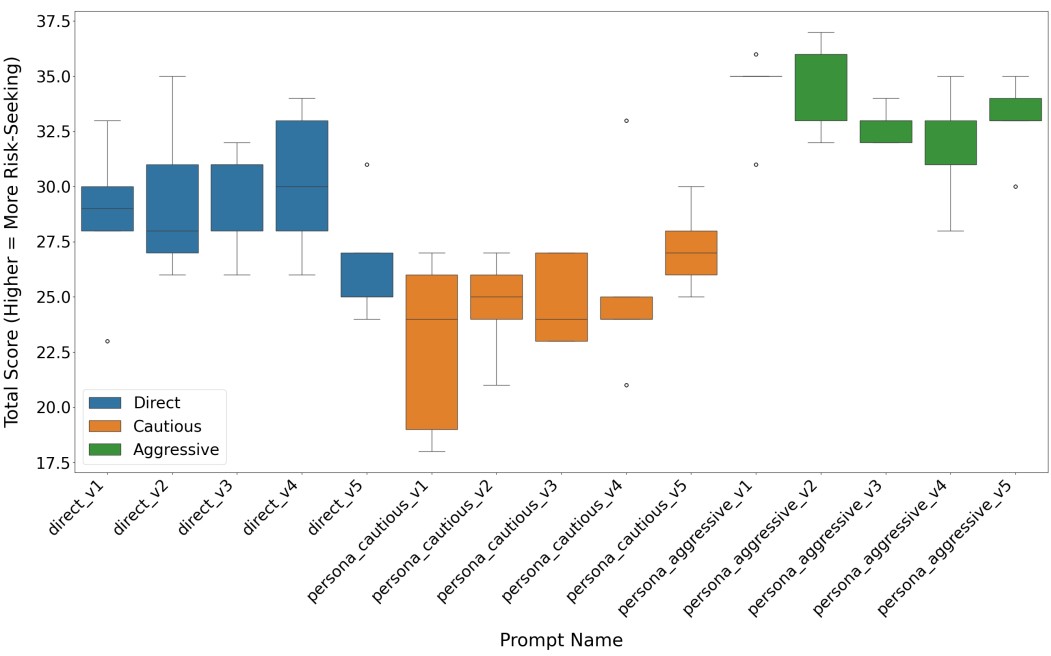

Figure 10: Distribution of Total Risk-Seeking Scores for Qwen2.5-MATH-7B across different prompt groups. Each prompt has 5 variants. Higher scores indicate greater risk-seeking behavior.

# D    MORE EXPERIMENT RESULTS AND DETAILS ON RISK PROFILING

Here we provide more details on the risk profiling experiments in Section 3.

## D.1    UTILITY FUNCTIONS

As noted in Section 3.1, we exhaust virtually all possible utility functions to fit the LLM's risk preference, aiming to ensure that we get a *good fit* before we proceed to analyze the LLM's risk preference using the fit as a proxy. As in the following, each utility function models and emphasizes certain aspect(s) of the risk attitude.

- **Linear Utility:** It assumes a full risk neutrality and there is no parameter.

$$u(x) = x$$

- **Power Utility:** A simple form for risk aversion ($\alpha < 1$) or risk-seeking ($\alpha > 1$).

$$u(x; \alpha) = x^\alpha$$

- **Quadratic Utility:** It implies increasing absolute risk aversion.

$$u(x; a, b) = ax - bx^2 \quad (b > 0)$$

- **Constant Relative Risk Aversion (CRRA):** It assumes that risk aversion is constant relative to wealth.

$$u(x; \gamma) = \begin{cases} \frac{x^{1-\gamma}-1}{1-\gamma} & \text{if } \gamma \neq 1 \\ \log(x) & \text{if } \gamma = 1 \end{cases}$$

- **Constant Absolute Risk Aversion (CARA):** It assumes risk aversion is constant regardless of wealth.

$$u(x; \alpha) = \begin{cases} \dfrac{1 - e^{-\alpha x}}{\alpha}, & \alpha > 0 \\ x, & \alpha = 0 \end{cases}$$

Note that we normalize the $x$ values by dividing them by 250 when using this utility function; otherwise, the exponential term becomes too small and can lead to numerical underflow.

- **Hyperbolic Absolute Risk Aversion (HARA):** A flexible function that includes CRRA and CARA as special cases.

$$u(x; a, b, \gamma) = \frac{1 - \gamma}{\gamma} (a + bx)^\gamma$$

- **Expo-Power Utility (Saha):** Another flexible form exhibiting both increasing and decreasing risk aversion.

$$u(x; \alpha, \theta) = \frac{1 - e^{-\alpha x^{1-\theta}}}{\alpha}$$

- **Prospect Theory Value Function:** It models loss aversion and diminishing sensitivity from a reference point $r_0$ (here, $r_0 = 0$).

$$v(x; \alpha, \beta, \lambda) = \begin{cases} (x - r_0)^\alpha & \text{if } x \geq r_0 \\ -\lambda(r_0 - x)^\beta & \text{if } x < r_0 \end{cases}$$

- **Epstein-Zin Utility:** A recursive utility function that disentangles risk aversion from the elasticity of intertemporal substitution ($\psi$).

$$U = \left[ (1 - \beta_d)c^{1-1/\psi} + \beta_d \left( E[V^{1-\alpha}] \right)^{\frac{1-1/\psi}{1-\alpha}} \right]^{\frac{1}{1-1/\psi}}$$

We implement the Epstein-Zin utility for a single lottery as a scalar-to-scalar mapping. Given a vector of lottery rewards $\mathbf{r} = (r_1, r_2, \ldots, r_n)$ and associated probabilities $\mathbf{p} =$

$(p_1, p_2, \ldots, p_n)$, along with parameters $\alpha$ (risk aversion), $\psi$ (intertemporal elasticity of substitution), and $\beta$ (discount factor), the utility is computed as follows.

First, we compute the weighted sum of the rewards raised to the power $(1-\alpha)$ and stabilize it with a small positive constant $\varepsilon$:

$$\text{exp\_term} = \sum_{i=1}^{n} p_i \, r_i^{1-\alpha}, \quad \text{exp\_term}_{\text{pos}} = \max(\text{exp\_term}, \varepsilon)$$

The inner term is then defined as

$$\text{inner} = \left(\text{exp\_term}_{\text{pos}}\right)^{\frac{1-1/\psi}{1-\alpha}}$$

Finally, the scalar Epstein-Zin utility is aggregated with the discount factor $\beta$ and a small constant $\varepsilon$ for numerical stability:

$$U = \left((1-\beta)\,\varepsilon^{1-1/\psi} + \beta\,\text{inner}\right)^{1/(1-1/\psi)}$$

Thus, for a single lottery, the function maps the scalar inputs (the weighted lottery rewards and parameters) to a single scalar output $U$.

- **Piecewise Power Utility (Friedman-Savage):** A three-part function designed to model agents who are risk-averse for small and large outcomes but risk-seeking for intermediate outcomes. The function is defined by two changepoints, $c_1$ and $c_2$, with different power exponents $(\alpha_1, \alpha_2, \alpha_3)$ in each region.

$$u(x) = \begin{cases} x^{\alpha_1} & \text{if } x < c_1 \\ y_1 + (x^{\alpha_2} - c_1^{\alpha_2}) & \text{if } c_1 \leq x < c_2 \\ y_2 + (x^{\alpha_3} - c_2^{\alpha_3}) & \text{if } x \geq c_2 \end{cases}$$

where $y_1$ and $y_2$ are constants ensuring the function is continuous. Typically, $\alpha_1, \alpha_3 < 1$ (concave/risk-averse) and $\alpha_2 > 1$ (convex/risk-seeking).

### D.2    BAYESIAN MODEL FITTING AND PRIORS

Here we provide more details on how the Bayesian fitting of the utility function in Section 3.2 works. We mainly rely on the Python implementation of MCMC in the package `PyMC`. For each candidate utility model, we place weakly informative priors on parameters to regularize estimation while reflecting conventional domain knowledge. All models were fitted with the No-U-Turn Sampler (NUTS), a Hamiltonian Monte Carlo algorithm, using the following sampling configuration:

`draws=3000, tune=1500, chains=6, cores=6, target_accept=0.97,`

and samples were returned as ArviZ `InferenceData` objects for subsequent summarization.

For the utility model in above, we use the following priors, respectively.

- Inverse-temperature / sensitivity: $\beta_{\text{sensitivity}} \sim \text{HalfNormal}(\sigma = 2.0)$, enforcing positivity and discouraging extreme determinism a priori.

- Scale / location priors: parameters denoted by "`a`" were given $\mathcal{N}(\mu = 1, \sigma = 1)$ priors; quadratic curvature terms "`b`" used $\text{HalfNormal}(\sigma = 1)$.

- Curvature / risk aversion parameters (e.g., "$\gamma, \alpha, \theta, \delta$"): $\text{HalfNormal}(\sigma = 1)$, encoding positivity while remaining weakly informative.

- Prospect-theory loss aversion: $\lambda \sim \mathcal{N}(\mu = 2, \sigma = 1)$.

- Epstein–Zin preference parameters: $\psi \sim \mathcal{N}(\mu = 1, \sigma = 0.5)$ and the discount factor $\beta_{\text{disc}} \sim \text{Beta}(2, 2)$.

- Probability-weighting models: e.g., Prelec's $\gamma$ and Gonzalez–Wu's $\delta, \gamma$ used $\mathcal{N}$ priors centered near typical literature values (see code).

- Piecewise (Friedman–Savage style) model:
  - Changepoint $c_1$ received a truncated normal prior: $c_1 \sim \mathcal{TN}(\mu = 0.25 \cdot M, \sigma = 0.10 \cdot M, \text{lower} = \varepsilon, \text{upper} = M)$, where $M$ is the maximum observed reward in the training data (so the prior is informed by the data scale), and $\varepsilon$ is a small positive constant.
  - The second changepoint was parameterized as $c_2 = c_1 + \delta$ with $\delta \sim \text{HalfNormal}(\sigma = 0.2 \cdot M)$ to enforce $c_2 > c_1$.
  - Curvatures for concave regions ($\alpha_1, \alpha_3$) used truncated normals concentrated below one (e.g., $\mu = 0.7$, truncated to $(\varepsilon, 1]$); the convex middle region parameter $\alpha_2$ was given a truncated normal with support $> 1$ (e.g., $\mu = 1.3$).
- For any parameter not explicitly listed above, we used weak Normal priors such as $\mathcal{N}(\mu = 0.8, \sigma = 0.5)$.

| Model / Parameter | mean | sd | hdi_3% | hdi_97% |
|---|---|---|---|---|
| **DeepSeek-R1-Distill-Llama-8B - SAHA Model** | | | | |
| $\alpha$ | 0.040 | 0.080 | 0.000 | 0.160 |
| $\theta$ | 0.592 | 0.174 | 0.299 | 0.929 |
| $\beta_{\text{sensitivity}}$ | 0.185 | 0.391 | 0.001 | 0.420 |
| **Llama-3.1-8B - Epstein-Zin Model** | | | | |
| $\alpha$ | 0.907 | 0.585 | 0.001 | 1.944 |
| $\psi$ | 1.059 | 0.514 | 0.203 | 1.852 |
| $\beta_{\text{disc}}$ | 0.482 | 0.188 | 0.118 | 0.868 |
| $\beta_{\text{sensitivity}}$ | 1.522 | 1.039 | 0.000 | 3.326 |
| **Llama-3.1-8B-Instruct - Epstein-Zin Model** | | | | |
| $\alpha$ | 3.533 | 0.058 | 3.422 | 3.641 |
| $\psi$ | 1.744 | 0.309 | 1.193 | 2.306 |
| $\beta_{\text{disc}}$ | 0.999 | 0.001 | 0.997 | 1.000 |
| $\beta_{\text{sensitivity}}$ | 55.568 | 0.884 | 53.959 | 57.283 |
| **Qwen2.5-7B - CRRA Model** | | | | |
| $\gamma$ | 0.660 | 0.264 | 0.278 | 1.186 |
| $\beta_{\text{sensitivity}}$ | 0.883 | 1.640 | 0.123 | 4.549 |
| **Qwen2.5-7B-Instruct - CRRA Model** | | | | |
| $\gamma$ | 0.564 | 0.051 | 0.460 | 0.617 |
| $\beta_{\text{sensitivity}}$ | 0.883 | 1.640 | 0.123 | 4.549 |
| **Qwen2.5-MATH-7B - Epstein-Zin Model** | | | | |
| $\alpha$ | 0.918 | 0.567 | 0.000 | 1.842 |
| $\psi$ | 0.949 | 0.447 | 0.089 | 1.690 |
| $\beta_{\text{disc}}$ | 0.521 | 0.215 | 0.089 | 0.883 |
| $\beta_{\text{sensitivity}}$ | 2.136 | 1.719 | 0.143 | 5.110 |

Table 4: Posterior summaries (mean, standard deviation, highest density interval) for all models

These priors (HalfNormal, TruncatedNormal, Beta, and Normal) accomplish two goals: they (i) encode mild prior beliefs (e.g., positivity of risk aversion parameters), and (ii) regularize estimation to improve sampler stability across models with varying parameterizations. We use data-aware priors for piecewise changepoints (based on $M$) to keep the change points in a plausible range given the reward scale. Model traces and posterior summaries are saved for each candidate utility (and probability-weighting) model; posterior means and credible intervals are used to compute predicted choice probabilities on a held-out test set and to compute accuracy metrics reported in the main paper. The relatively tight `target_accept`=0.97 and the moderate number of tuning draws were chosen to reduce divergence risk for some of the more complicated models (e.g., Epstein–Zin and piecewise specifications). Standard MCMC diagnostics (R-hat divergence checks) were examined for each trace; models exhibiting pathologies were inspected and recorded as N/A in the tables.

The fitted parameters and the related distributional properties are summarized in Table 4.

### D.3 MORE RESULTS

Table 5 and Table 6 extend the results in Figure 2 and give a more comprehensive report of the utility function fitting results. The N/A entries mean the parameter learning doesn't converge to a solution.

The two tables correspond to the two datasets where the expected utilities of the two options are the same or different. We can see that the case where the expectations are different gives a better fitting performance. This indicates that this dataset is more informative and effective in eliciting the risk preference of the LLMs.

| Model | Power | CRRA | CARA | HARA | SAHA | Prospect | Epstein-Zin |
|---|---|---|---|---|---|---|---|
| Llama-3.1-8B | 50.80 | 52.28 | 51.36 | N/A | 51.64 | 50.04 | 69.60 |
| Qwen2.5-7B | 70.96 | 74.86 | 71.84 | 29.56 | 72.36 | 71.72 | 69.40 |
| Llama-3.1-8B-Instruct | 55.76 | 55.76 | 55.84 | 56.20 | 55.4 | 56.00 | 93.88 |
| Qwen2.5-7B-Instruct | 70.96 | 82.72 | 71.84 | 29.56 | 72.36 | 71.72 | 69.40 |
| Qwen2.5-MATH-7B | N/A | 57.12 | 57.24 | 42.60 | 57.52 | N/A | 73.68 |
| DeepSeek-R1-Distill-Llama-8B | N/A | 53.16 | 53.16 | N/A | 60.12 | N/A | 31.20 |

Table 5: Fitted Utility Function Accuracy Comparison (%) (Different expectation dataset)

| Model | Power | CRRA | CARA | HARA | SAHA | Prospect | Epstein-Zin |
|---|---|---|---|---|---|---|---|
| Llama-3.1-8B | 53.24 | 52.52 | 48.20 | N/A | 53.36 | 52.08 | 63.88 |
| Qwen2.5-7B | 42.48 | 68.56 | 57.92 | 58.72 | 58.72 | 42.44 | 64.76 |
| Llama-3.1-8B-Instruct | 50.12 | 55.48 | 20.12 | 50.00 | 52.40 | 49.76 | 82.98 |
| Qwen2.5-7B-Instruct | 42.48 | 68.56 | 57.92 | 58.72 | 58.56 | 62.44 | 44.76 |
| Qwen2.5-MATH-7B | N/A | 46.32 | 72.36 | 46.48 | 52.80 | N/A | 64.08 |
| DeepSeek-R1-Distill-Llama-8B | N/A | 48.84 | 33.88 | N/A | 55.64 | N/A | 31.6 |

Table 6: Fitted Utility Function Accuracy Comparison (%) (Same expectation dataset)

# E   MORE EXPERIMENT RESULTS AND DETAILS ON RISK MODULATION

## E.1   IN-CONTEXT PROMPTING

For the numerical experiment of Figure 3, we use the following prompt for in-context learning. We sample $k$ example questions and the test question randomly without replacement from the sample dataset $\mathcal{D}_{\text{align}}$, and then generate the prompt below.

---

**In-Context Prompt Example**

You are a decision-making assistant. Follow the examples' risk attitude, try to understand their decision logics, and choose the option (A or B) for the test question in the end.
Here are some examples:
Question: [Example question 1]
Choice: [A or B]
Question: [Example question 2]
Choice: [A or B]
...
Question: [Example question k]
Choice: [A or B]
Now predict the choice for the next question:
Question: [Test question]
Choice:

---

## E.2   SFT & DPO EXPERIMENT DETAILS

The SFT experiments are conducted on six pre-trained language models (Llama-3.1-8B, Qwen2.5-7B, Llama-3.1-8B-Instruct, Qwen2.5-7B-Instruct, Qwen2.5-MATH-7B, DeepSeek-R1-Distill-Llama-8B) using a fixed-utility ground truth. Base models are downloaded from Hugging Face and loaded via `transformers`, then quantized to 4-bit precision using `BitsAndBytesConfig`.

The SFT training dataset is constructed as follows. Each data sample in $\mathcal{D}_{\text{align}}$ contains two options (labeled A and B), each with two possible rewards and associated probabilities, and the label $y_i$ is

generated following the probability rule (3). Each data sample is then converted into an SFT sample with the following template:

> **SFT template**
>
> You are an economic decision-making agent. Analyze the options and reply with your choice as a single letter: A or B.
> Question: [description of options]
> Answer:

The completion is simply the preferred choice, "A" or "B", derived from the utility calculation. This dataset, consisting of `prompt` and `completion` columns, is then used to train the language model via `SFTConfig` and `SFTTrainer`. Parameter-efficient fine-tuning (LoRA) is applied with rank $r = 16$, $\alpha = 32$, and dropout 0.05, targeting the attention and projection layers (`q_proj`, `k_proj`, `v_proj`, `o_proj`, `gate_proj`, `up_proj`, `down_proj`) without additional bias. The training uses a per-device batch size of 2, gradient accumulation of 8 steps, learning rate $5 \times 10^{-6}$, and 2 epochs. A mixed precision (FP16) is enabled, with model checkpoints saved every 500 steps. Evaluation accuracy is measured as the fraction of predictions matching the ground truth. Utility function parameters (e.g., $\alpha, \beta, \theta, \gamma, b$, reference point) are configurable via command-line arguments, allowing flexible testing across different models and utility functions.

DPO fine-tuning shares many implementation details with SFT, such as using the same base models, LoRA adapters, mixed-precision training, and the same utility-based preference data. The key difference lies in the training objective: instead of using a fixed ground-truth label for each prompt, DPO leverages pairwise preferences between two possible completions (here, choices A and B for a given lottery). For each prompt, the DPO dataset contains three key columns:

- `prompt`: the natural language description of the lottery options (same template as SFT),
- `chosen`: the option with higher utility,
- `rejected`: the alternative option.

The model is trained using a pairwise loss to assign higher likelihood to the `chosen` completion relative to the `rejected` completion. We use `DPOConfig` and `DPOTrainer` for the training process, keeping similar hyperparameters as SFT (LoRA rank $r = 16$, $\alpha = 32$, dropout 0.05, per-device batch size 2, gradient accumulation 8, learning rate $5 \times 10^{-6}$, 2 epochs, FP16, checkpoint every 500 steps).

### E.3 OUT-OF-DISTRIBUTION EXPERIMENTS

#### E.3.1 FOUR-OPTION LOTTERY

The questions in this setting are generated using the same method described in Section B.4. However, instead of producing two options, we construct four different options for each question.

We then use the post-trained models to generate answers to these questions. Each generated answer is compared against the ground-truth option, which is computed using the utility function and parameters employed in training the DPO model. Finally, we evaluate model performance based on accuracy over this four-choice dataset.

#### E.3.2 DOSPERT DATASET

We evaluated the behavior of post-trained large language models on the DOSPERT questionnaire (Blais & Weber, 2006), which consists of 30 questions across five domains: *Ethical*, *Financial*, *Health/Safety*, *Recreational*, and *Social*. Each question is evaluated on a 7-point Likert scale, where a score of 1 indicates "extremely unlikely" and a score of 7 indicates "extremely likely." For each DOSPERT item, for example, "Taking some questionable deductions on your income tax return", the agent is asked to respond along two dimensions:

- Risk Taking: "How likely are you to engage in this activity?" This measures the extent to which the agent would take the action.

- Risk Perception: "How risky do you perceive this activity to be?" This measures how risky the agent perceives the proposed action.

We tested a base language model (Qwen2.5-7B) alongside several DPO fine-tuned variants with different utility functions: (i) CRRA ($\theta = -5$); (ii) CRRA ($\theta = 0.71$); (iii) CRRA ($\theta = 1$). All fine-tuned models were implemented using LoRA adapters on top of the base model.

**Evaluation**. The evaluation followed these steps:

(a) Prompting: For each DOSPERT item, we constructed prompts separately for the two dimensions: risk-taking likelihood and risk perception. The exact prompt for risk-taking was:

> **Risk-Taking Prompt**
>
> On a scale of 1 (extremely unlikely) to 7 (extremely likely), how likely are you to engage in the following activity?
> Answer with a single number between 1 and 7.
> Question: Taking some questionable deductions on your income tax return.
> Answer:

> **Risk-Perception Prompt**
>
> On a scale of 1 (not at all risky) to 7 (extremely risky), how risky do you perceive the following activity to be?
> Answer with a single number between 1 and 7.
> Question: Taking some questionable deductions on your income tax return.
> Answer:

The model was configured with **temperature = 0.7** and **top-p = 0.9** to allow moderate randomness while keeping responses consistent. A maximum of 50 tokens was generated per query.

(b) Answer Extraction: The model's output was parsed using regular expressions to extract the Likert score. The regex handled different formatting cases, such as:

- Numbers enclosed in parentheses, e.g., "(3)".

- Numbers preceded by text, e.g., "a score of 4" or "would be a 5".

- Ranges or extra text such as "1 (extremely unlikely) to 7 (extremely likely)" were removed.

The extraction regex first attempted to match patterns like `\(([1-7])\)` or `\b([1-7])\b`. If no valid number was found, the query was retried up to three times to ensure valid data.

(c) Repetition and Averaging: To mitigate stochasticity in the model's responses, each question was queried **10 times**, and the valid responses were averaged to produce a single Likert score for each item.

(d) Aggregation: For each DOSPERT domain (Ethical, Financial, Health/Safety, Recreational, Social), the averaged scores across all questions in that domain were computed, resulting in domain-level mean scores for both risk-taking and risk perception. These domain-level averages were then used for visualization (radar plots) and cross-model comparisons.

We show more DOSPORT items across different domains below.

### Ethical Questions

1. Taking some questionable deductions on your income tax return.
2. Having an affair with a married man/woman.
3. Passing off somebody else's work as your own.
4. Revealing a friend's secret to someone else.
5. Leaving your young children alone at home while running an errand.
6. Not returning a wallet you found that contains $200.

### Financial Questions

1. Investing 5% of your annual income in a very speculative stock.
2. Investing 10% of your annual income in a moderate growth mutual fund.
3. Investing 10% of your annual income in a new business venture.
4. Betting a day's income at the horse races.
5. Betting a day's income on the outcome of a sporting event.
6. Betting a day's income at a high-stake poker game.

### Health/Safety Questions

1. Drinking heavily at a social function.
2. Engaging in unprotected sex.
3. Driving a car without wearing a seatbelt.
4. Riding a motorcycle without a helmet.
5. Sunbathing without sunscreen.
6. Walking home alone at night in an unsafe area of town.

### Ethical Questions

1. Taking some questionable deductions on your income tax return.
2. Having an affair with a married man/woman.
3. Passing off somebody else's work as your own.
4. Revealing a friend's secret to someone else.
5. Leaving your young children alone at home while running an errand.
6. Not returning a wallet you found that contains $200.

### Social Questions

1. Admitting that your tastes are different from those of a friend.
2. Disagreeing with an authority figure on a major issue.
3. Choosing a career that you truly enjoy over a more prestigious one.
4. Speaking your mind about an unpopular issue in a meeting at work.
5. Moving to a city far away from your extended family.
6. Starting a new career in your mid-thirties.

### E.4 RETURNING TO GRABLE & LYTTON RISK TOLERANCE SCALE

After training the models using DPO, with data from the CRRA utility function with parameters $\theta = -5$, $0.71$, and $1$, we re-evaluate the models on the Grable & Lytton dataset, and see how it changes comparing to Table 1. We employ the same direct prompt with five variations setting, and set the generation temperature to $0.7$.

Our results indicate that compared to Table 1, models trained with a risk-seeking parameter ($\theta = -5$) tend to exhibit higher risk-taking behaviors compared to the original model. Conversely, models trained with a risk-averse parameter demonstrate more conservative decision-making patterns.

Table 7: Post-training effect of DPO on different parameters evaluated with CRRA utility function. Scores indicate the total score / risk level.

| Model | CRRA ($\theta = -5$) | CRRA ($\theta = 0.71$) | CRRA ($\theta = 1$) |
|---|---|---|---|
| Llama-3.1-8B | 30.92 / Above-average | 26.28 / Average | 26.14 / Average |
| Qwen2.5-7B | 29.56 / Above-average | 26.66 / Average | 26.00 / Average |
| Llama-3.1-8B-Instruct | 28.92 / Above-average | 25.20 / Average | 24.92 / Average |
| Qwen2.5-7B-Instruct | 28.14 / Above-average | 23.56 / Average | 22.46 / Below-average |
| Qwen2.5-MATH-7B | 31.08 / Above-average | 28.16 / Average | 27.48 / Average |
| DeepSeek-R1-Distill-Llama-8B | 28.60 / Above-average | 23.54 / Average | 23.56 / Average |

