# OpenReview forum: "Risk Profiling and Modulation for LLMs"
_ICLR.cc/2026/Conference — ICLR 2026 Conference Withdrawn Submission_

### Official Review · Reviewer_G6XQ · 2025-10-20

**Soundness:** 3
**Presentation:** 3
**Contribution:** 2
**Rating:** 4
**Confidence:** 5

**Summary:**

This paper studies risk profiles in pretrained, instruction-tuned, and RLHF-aligned open-weight models. First, they use a well-known risk tolerance scale and then generate their own dataset simulating lottery choices to study the fit of different known utility functions. Later, they explore the risk modulation of these LLMs, i.e., the ability to steer their behavior towards certain utility functions. Their results show better results for instruction-tuned models and better performance with SFT/DPO with respect to prompting; while the modulation results don’t generalize to domains that are quite different from their lottery dataset.

**Strengths:**

- The paper deals with an important problem, where we need to understand how LLMs make decisions when it comes to risk (e.g., seeking or aversion).
- I appreciate how the authors didn’t stop at just prompting or using SFT/DPO in their own domain. My main concern would’ve been the generalization beyond this domain, and they got ahead.
- The writing of the paper is quite clear, and I found all the formalizations very useful.
- There are a lot of details in the paper and appendix, which highlight its reproducibility.

**Weaknesses:**

**My main issue with the paper is the relevance of its contribution**. Otherwise, what the authors do in the paper, they do it well. The contributions I see are:

(a) studying risk profiles by prompting with a known risk scale questionnaire

(b) studying risk profiles by generating a controlled dataset and testing utility functions

(c) risk modulation via prompting, SFT, and DPO

(d) robustness test

There are too many papers to count where LLMs are studied with simple questionnaires, such as in (a). However, these questionnaires are often up for debate in the behavioral sciences. I think it’s good that the authors included this simple approach in the paper, but it’s not a significant contribution by itself.

When it comes to (b), the dataset generated is very simple, containing binary choices (later more) for a very specific problem. It already makes me question whether what we learn can generalize to some useful scenario (the “robustness test” reveals that it might not). Studying whether these models have learned known utility functions is quite interesting, but I see a missed opportunity for doing something like symbolic regression to really extract a utility function. The accuracy itself falls short, since it could show overfitting for this particular task (as it’s revealed in the “robustness test”).

For (c), these results are quite common, and it is expected to see prompting < SFT < DPO. I’m not sure we have learned anything new from these experiments, except that the common pattern also applies to risk profiles.

Finally, (d) reveals that models that can be modulated well don’t show the same behavior in different domains. To me, this looks like these methods force models to learn a particular task, but have nothing to do with true behavior.

### Minor Details (These didn’t affect my decision)

While the **writing** is perfectly clear, the introduction reads like a simple related-work section. I didn’t find that very exciting, and I think it’s a missed opportunity for explaining why this problem matters. The paper gets more interesting in Section 3, but that’s almost half of the main paper.
Several papers would be worth including. For example, sentences like this one

> While their reasoning and language generation abilities are well established, recent work has begun to probe whether LLMs exhibit systematic behavioral tendencies similar to those documented in human decision-making

Could reference other recent work (these are a few examples)

- Liu, R., Sumers, T. R., Dasgupta, I., & Griffiths, T. L. (2024). How do large language models navigate conflicts between honesty and helpfulness?. arXiv preprint arXiv:2402.07282.
- Liu, R., Geng, J., Wu, A. J., Sucholutsky, I., Lombrozo, T., & Griffiths, T. L. (2024). Mind your step (by step): Chain-of-thought can reduce performance on tasks where thinking makes humans worse. arXiv preprint arXiv:2410.21333.
- Cherep, M., Maes, P., & Singh, N. (2025). LLM Agents Are Hypersensitive to Nudges. arXiv preprint arXiv:2505.11584.
- Xie, C., Chen, C., Jia, F., Ye, Z., Lai, S., Shu, K., ... & Li, G. (2024). Can large language model agents simulate human trust behavior?. Advances in neural information processing systems, 37, 15674-15729.
- Dettki, H. M., Lake, B. M., Wu, C. M., & Rehder, B. (2025). Do Large Language Models Reason Causally Like Us? Even Better?. arXiv preprint arXiv:2502.10215.

**Questions:**

- Did you consider running experiments with closed models? I’m aware of why you use open-weight models, but I’d be curious to see partial results for SOTA models. Note: This has nothing to do with my rating, and I don’t expect you to collect more results.
- I might be missing something in Figure 3 (right) and Figure 2 (left). The table shows how Qwen2.5-7B-Instruct is best fitted by CRRA, but then the modulation by the same utility function is poor.
- For the questionnaire, do you randomize the order of the questions as well? I can see in the Appendix that you do it for your generated dataset, but I’m curious about the questionnaire given LLMs’ sensitivity to the order of multiple-choice questions.
- Suggestion 1: Figure 3 (and the similar ones in the appendix) could have a line for chance.
- Suggestion 2: Figure 3 (and others) could show the average over utility functions and have one plot with all the models, and include all of them by function in the appendix.
- Suggestion 3: Figure 2 (right) is quite confusing considering the caption: “Visualization of the three best-fitted functions.” I see you have to ignore Epstein-Ziv, but that means you don’t actually plot the best-fitted.

---

### Official Review · Reviewer_oVUw · 2025-10-31

**Soundness:** 2
**Presentation:** 1
**Contribution:** 1
**Rating:** 2
**Confidence:** 4

**Summary:**

This paper introduces a novel pipeline for eliciting, steering, and modulating the risk profiles of large language models (LLMs), drawing on concepts from behavioral economics and finance. It effectively characterizes how risk profiles vary across different model classes and training stages, and demonstrates how post-training interventions reshape these profiles. While the experimental findings are meaningful and shed critical light on the inherent risks present during large model training, the core methodology does not introduce significant technical innovation.

**Strengths:**

1. This paper reveals the inherent risk patterns across different categories and training phases of LLMs through extensive experiments, and proposes methods for modulating the risks of LLMs.

**Weaknesses:**

1. The main text does not provide an explanation for Figure 1 and Tabel 3, making it difficult for readers to understand the context and significance of the illustration.
2. Although the work attempts to integrate Bayesian learning and utility theory into the domain of LLMs, the proposed methodology lacks significant originality and does not present substantial technical advancements over existing approaches.
3. All the models used in the experiments are either 7B or 8B in size. The authors are encouraged to investigate and report on how varying model scales impact the experimental results.
4. In Figure 4, the domain-level scores of different DPO fine-tuned models seem to be in a trade-off with the base model's performance, making it challenging to intuitively assess the effectiveness of the proposed methods. The authors are encouraged to provide a clearer interpretation of these experimental results and to include additional experiments with other fine-tuned models for more robust validation.
5. The references are inconsistent. Please check and standardize them.

**Questions:**

1. Why are the results of Bayesian fitting of the utility functions for all models not presented on the right side of Figure 2?
2. Please refer to the “Weakness” section for related questions.

---

### Official Review · Reviewer_ZBw3 · 2025-10-31

**Soundness:** 3
**Presentation:** 3
**Contribution:** 2
**Rating:** 4
**Confidence:** 5

**Summary:**

This paper
- introduces a pipeline to measure and then steer the risk preferences of LLMs using tools from behavioral econ
- finds that instruction-tuned models align well with standard utility formulations, while pre-trained and RLHF-aligned models show more complex or inconsistent decision patterns
- finds that in-context prompting is unstable and ineffective for steering LLMs towards having particular preferences; SFT and DPO work much better for this purpose (especially the latter, as measured in OOD tasks)

**Strengths:**

- timely topic: agentic LLMs are increasingly used for decision-making under uncertainty in high-stakes domains, and it's important that their risk preferences much that of humans (classic principal-agent problem)
- authors provide an OOD evaluation of their tuned models
- the authors use accepted tools from the behavioral econ literature, like the Grable & Lytton scale

**Weaknesses:**

I think this paper is missing the forest for the free along several axes:
- LLMs are not good at doing arithmetic reasoning out of the box, the kind of which is needed to adequately compare two different risky outcomes. There are no ablations done here, for example, to determine how important tokenization is to the LLMs' observed behavior. What if the outcomes were for larger numbers (e.g., 100K)? What if you included both dollars and cents in the outcome? I suspect that this would dramatically change the results.
- How well do these risk preferences transfer to non-monetary domains in which LLMs make decisions? Related to the previous point, what if you framed this not as a lottery but as choosing between two products instead (a more realistic scenario)?
- It is pretty well known at this point that most state-of-the-art post-training objectives are themselves prospect-theoretic (see: the KTO paper). Does this help explain why DPO works well? Does KTO --- which is explicitly based on a prospect theoretic model of utility (over implicit rewards, not dollars) --- induce the model to have loss/risk aversion when it comes to choosing among lotteries?

To me, these are the far more interesting research questions. The paper is its current form is not unsound, but fairly unremarkable.

**Questions:**

See above.

---

### Note · Authors · 2025-11-27

I have read and agree with the venue's withdrawal policy on behalf of myself and my co-authors.